# A Descriptive Study of Webpage Designs for Posting Privacy Policies for Different-Sized US Hospitals to Create an Assessment Framework

Karen Schnell *, Kaushik Roy and Madhuri Siddula 

Department of Computer Science, North Carolina Agricultural and Technical State University, Greensboro, NC 27411, USA; kroy@ncat.edu (K.R.); msiddula@ncat.edu (M.S.)
* Correspondence: klschnell@ncat.edu

**Abstract:** In the United States, there are laws and standards guiding how people should be informed about the use of their private data. However, the challenge of communicating these guidelines to the naïve user is still at its peak. Research has shown that the willingness to read privacy statements is influenced by attitudes toward privacy risks and privacy benefits. Many websites publish privacy policies somewhere on their web pages, and it can be difficult to navigate to them. In the healthcare field, research has found that health information websites' key information is presented poorly and inconsistently. For the policies to be legally binding, a person must be able to find them. In the healthcare industry, where sensitive data are being collected, research on how a user navigates to privacy policies for different size hospital websites is limited. Studies exist about privacy policies or website design and not both. This descriptive study involved ascertaining commonalities and differences among different-sized hospitals' website designs for supporting privacy policies. A foundation framework was created using Web Content Accessibility Guidelines (WGAC) principles and the literature review findings for evaluating practices for website publishing of privacy policies. The results demonstrated a very low variance in the website design concepts employed by hospitals to publish their privacy policy.

**Keywords:** privacy; website design; healthcare

## 1. Introduction

This descriptive study focuses on US hospital website design and the ability to locate the privacy policy. Websites are often used for privacy rights communication. Communication with individuals regarding their privacy rights is important. According to a survey study of university students by Sigmond [1], the willingness to read privacy statements is influenced by attitudes towards privacy risks and privacy benefits. Berland et al. [2] studied English and Spanish health information websites and found that key information is poor and inconsistent. Web Content Accessibility Guidelines (WGAC) [3] principles provide a framework for evaluating practices for website publishing. In a study by Page et al. [4] on patients' opinions on privacy, consent, and the disclosure of health information, most of the surveyed groups identified privacy and control over personal health information were important. The legal guidelines for protecting and informing patients of their rights is the Health Insurance Portability and Accountability Act (HIPAA) [5] in healthcare settings. HIPAA is used to make patients aware of how their Personally Identifiable Information (PII) data are going to be used, shared, and processed and the length of time that it will be retained. In research, studies have been published on the design of privacy policies or the accessibility design of websites but not both. In healthcare, there is an opportunity to study hospital webpage design for posting privacy notices. This study investigates various hospital website design practices to develop a foundation framework using WGAC principles for assessment to identify techniques in publishing privacy policies.

The main contributions of this study are as follows: (1) The literature review revealed the gap in research on website design in healthcare to attract users to locate privacy policies. To date, published research has separately investigated website designs or privacy policies. Studies have not explored how to ensure that a user locates the privacy policy online and what techniques can improve their ability to find it. (2) Based upon select WGAC principles, a framework for assessing website design to support privacy policy posting was implemented to compare the website design of different-sized hospitals. This structure provides a foundational standard methodology to evaluate design practices for online privacy policy publication. For future research, this provides a baseline to refine and enhance the development of a tool that could be used to appraise website designs that improve the user's experience in locating privacy policies. (3) It reveals future research opportunities to expand the use of a privacy policy posting standardized website design to other industries. Further, the homogeneous website design methods that were widely used by hospitals can be researched to find which are more beneficial to use to obtain improved user reviews of their privacy policies over others.

The paper is organized as follows—(1) Introduction to the problem space; (2) background on healthcare industry website design for privacy protection, (3) proposed foundational framework assessment of website posting of privacy policies; (4) methodology; (5) results; (6) discussion; (7) limitations; (8) future research; and (9) conclusion.

## 2. Background

### 2.1. Healthcare Websites

Hospital websites are important for patients to learn about where to obtain healthcare. Alajarmah [6] performed a study that looked at the accessibility design of twenty-five countries' public healthcare websites. It was learned that most websites had accessibility barriers in terms of perception of information and operability of user interface items. A key finding was the need for training in Web accessibility requirements and universal design principles. In a study by Saghaeiannejad-Isfahani et al. [7] using the WebMedQual scale, which is a checklist to evaluate the quality of medical websites across eight main components, the lowest average score was in privacy. Rangraz Jeddi et al. [8] assessed several evaluation models for ascertaining the quality of hospital websites. Essential criteria for evaluation include accessibility, content, design, security, and confidentiality of personal information. Alhuwail et al. [9] used four dimensions for evaluating Kuwait hospital websites. These included accessibility, usability, presence, and content. From the investigation of nine hospitals, they found that the websites primarily promoted services rather than engaged with patients. In a study of Health Centers, Calvano et al. [10] assessed website usability, which equates to the ease of user experience when engaged with the website. Usability has four categories: accessibility, marketing, content quality, and technology. Content had the highest score, and technology had the lowest score.

### 2.2. Website Design

To help guide the design and development of a privacy notice and website, one can "google" or search the Internet and locate templates. Moreover, there are blogs, YouTube videos, articles, and books. For designing websites to support persons with disabilities, the Web Content Accessibility Guidelines (WCAG) provides "a single shared standard for web content accessibility that meets the needs of individuals, organizations, and governments internationally". With a focus on making web content more accessible, the guide provides information and recommendations on content design for text, images, sounds, code, structures, presentation, etc. In a research study by Cline et al. [11], they looked at design criteria for evaluating health information websites. The design feature criteria included: (1) accessibility; (2) ease of use; (3) links between sites; and (4) aesthetic and format characteristics such as text, audio, and visual style format. They also cited that there are very few systematic mechanisms for rating and standards for evaluating websites.

### 2.3. Privacy Policy Website Hosting

There is no legal mandate to host legal agreements online. However, to have them readily available for "any time" access for a subject to read, putting them on a business website is beneficial. Legal documents are required by law to be reachable and readable. Moreover, by law, they must be clearly associated with the business entity. Hosting an institution's Privacy Policy on its website is a preferred option to allow access. Having a downloadable version is helpful but not a legal obligation. Some companies have considered using a hosting service like GitHub to access legal documents. In the US, the Ninth Circuit case Nguyen v. Barnes & Noble [12] ruled that users must have actual notice of the Terms of Service and Privacy Policy. If they cannot access this information so they can agree to the conditions, then the legality of the terms may not be enforceable. Having a Privacy Policy is mandatory in many countries like the United States, Australia, the United Kingdom, Canada, and the European Union. The Federal Trade Commission requires that a Privacy Policy be conspicuous and "reasonably accessible". Therefore, to find a policy, the location and accessibility matter to the users.

### 2.4. Issues with Privacy Notices

Privacy notices are usually not read. According to a survey study of university students by Sigmond [1], the willingness to read privacy statements is influenced by attitudes towards privacy risks and privacy benefits. Social norms, understanding the content, and willingness to spend time and effort to read were key factors in whether the policy was reviewed or not.

Other issues with privacy notices are transparency and trust. Gerl et al. [13] highlighted that privacy notices lack transparency because of high complexity and missing details. In healthcare, the lack of transparency is amplified by the integration of departments, third parties, and services sharing data. They proposed the use of a Layered Privacy Language (LPL) to manage the privacy policy requirements, consent, and presentation to patients. This approach would personalize the privacy notification experience with a process and design. In terms of trust, Ruotsalainen et al. [14] cited in their work on integrated healthcare systems that ethics, privacy, and trust challenges exist. Their study objective was to develop a system that included computational trust models to enforce privacy with computer-understandable policies. Trust would be increased with the ability of service providers to customize their policies.

Schaub et al. [15] examined "why" privacy notices failed to inform end users of their rights. They found that the design process needs to include timing as to when the notice should be presented to the user, how it is delivered, how it is displayed or communicated, and what end-user choices are integrated into the notice. Further, they identified the following influences on the design of privacy notices: (1) conflicting purposes for end users because they should be transparent on the processing and gathering of their data while companies see a legal obligation and the need to avoid lawsuits, (2) they lack designs that allow for choices for users to opt-in or opt-out, (3) it takes a long time to read and comprehend the terminology found in the body of notifications, and (4) the location of the notice is usually not readily accessible, i.e., having to go to a website, following menus, contacting the business. Karegar et al. [16] measured user engagement using different formats than the normal text found in notices. They used swipe, drag, and drop and checkboxes. Their study revealed that creating different ways to interact with the form could increase user attention to sections of it. To increase user attention to review a privacy policy, Tabassum et al. [17] used the novel approach of a comic-based interface. Their results showed that user attention was held longer than in text-based formats.

### 2.5. Privacy Policy Publishing Assessments

Health information technologies (HIT) are important for providing users with information. This includes websites. LaMonica et al. [18] performed research that identified a growing concern about understanding privacy risks associated with HIT and the confusing



nature of privacy policies. The study applied a 23-item privacy policy risk assessment tool to assess whether HITs supported recommended privacy policy standards. Results found that users "wanted privacy information to be easily accessible, transparent, and user-friendly". Zimmeck et al. [19] developed a proof-of-concept browser extension that analyzed privacy policies for essential policy terms. The Privee tool architecture was based on the notice-and-choice principle for locating and understanding privacy policies on websites. It increased understanding and transparency by extracting key terms from privacy policies. Johnson et al. [20] researched templates for authoring security and privacy policies. The framework used new and existing policy guidelines to develop standardized systems. These assessments and frameworks focused on privacy policies and content rather than including design concepts of the website. These studies highlight the limited research on designing websites for locating the privacy policy.

## 3. Proposed Foundation Framework for Assessment of Website Design for Posting Privacy Policies

Schnell et al. [21] performed a study on the use of WCAG checkpoints and the design of higher education and financial institution websites for the visually impaired to locate a privacy policy. The study used compliance to six selected WCAG checkpoints for website assessment. Predictability issues included the number of URLs to click, the name of the entry title of the URL location, and the size of the title font. Readability issues included content spanning several tabs and the need to visit multiple URLs to be able to review the notice fully. Using these learnings, this study expands to the healthcare industry and proposes a set of questions to develop a reusable assessment framework. The following questions were developed to assess different-sized US hospitals' website designs for posting a privacy policy:

1.   Is the initial entry to the privacy notice located on the first page of the hospital website?
2.   Where on the website page is the initial entry to the privacy notice located?
3.   What is/are the terms used for the title for the initial entry to the privacy notice?
4.   How many URL clicks does it take to get to the privacy notice?
5.   Is any attention-eliciting website design elements used, as cited in research [16,17], such as drag, dropdowns, swipe, animation, or checkboxes?
6.   Are there website design differences amongst hospitals of varying sizes?

## 4. Methodology

One hundred hospital websites were used based on the list of the "2021 Fortune/IBM Health 100 Top Hospitals" [22]. The websites were manually reviewed for the following criteria: (1) Initial entry to the privacy notice, (2) location of the initial entry to the notice on the webpage, (3) capture of the initial entry title to the policy, (4) the count of URL clicks to the detailed privacy notice or statement (Figure 1), and (5) capture of the use of any of the following attention eliciting website design elements: drag, dropdowns, swipe, animation, or checkboxes. Manual review was chosen since no tools exist to perform such an assessment. Automating would require creating different scripts for parsing the unstructured text. Depending on the website's front-end design, there are challenges in creating a script to record the depth of the URLs to locate the privacy policy. The policy could be a PDF, HTML code, or various other applications that can be used. Comparisons among the four categories of hospitals were performed (Table 1).

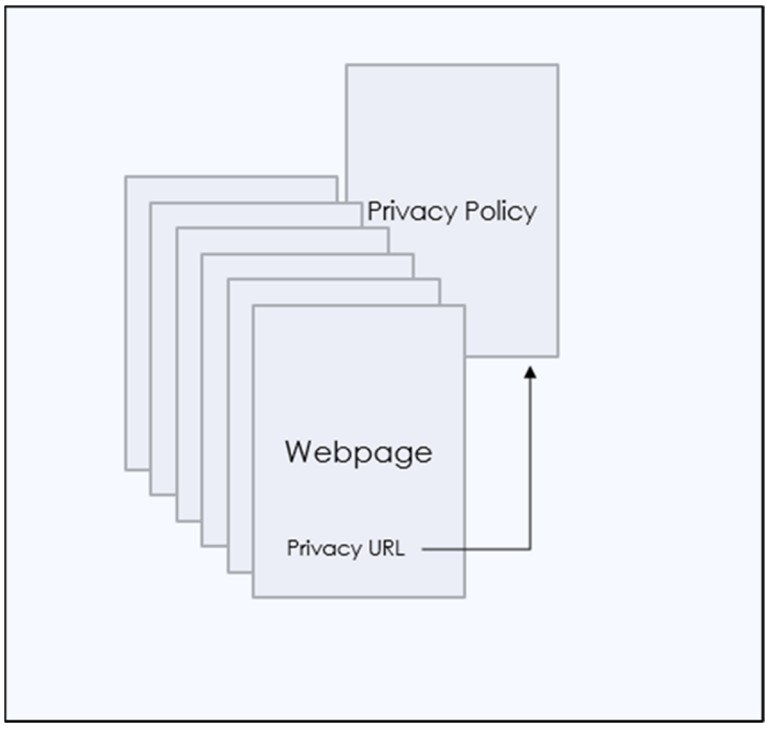

**Figure 1.** URL to Privacy Policy from main webpage.

**Table 1.** Criteria comparison matrix example.

| Assessment Framework | Major Teaching | Large Community | Medium Community | Small Community |
|:---:|:---:|:---:|:---:|:---:|
| Initial entry to the privacy notice | | | | |
| Location of the initial entry to the notice on the webpage | | | | |
| Capture of the initial entry title to the policy | | | | |
| Count of URL clicks to the detailed privacy notice or statement | | | | |
| Use of any of the following attention-eliciting website design elements | | | | |

### 4.1. Sample Selection

Belanger [22] performed a study to identify the top one hundred hospitals in the US for IBM Watson Health. The researchers evaluated 2675 short-term, acute care, non-federal US hospitals. So federally funded healthcare institutions such as the veteran's administration, military hospitals, and federal prisons or mental health facilities were not included. The research was based upon public data sets from (1) Medicare cost reports, (2) Medicare Provider Analysis and Review (MEDPAR) data, and core measures and patient satisfaction data from the Centers for Medicare and Medicaid Services (CMS) Hospital Compare website. The researchers clearly noted that no awards or payments are used for selection. This sample set was selected with standard criteria and varied primarily in facility size.

Based on the hospital bed size and teaching status, the following class definitions were derived:

- Major Teaching—208 hospitals
- Teaching—605 hospitals
- Large Community—254 hospitals
- Medium Community—774 hospitals

- Small Community—834 hospitals

For determination of the top hospitals, the researchers used a survey that measured the following domains: (1) clinical outcomes such as mortality, complications, infections, and readmissions; (2) operational efficiencies such as emergency room throughput, length of stay, Medicare spending, and cost per discharge; (3) patient experience as measured by overall patient rating from the Hospital Consumer Assessment of Healthcare Providers and Systems (HCAHPS) survey tool; (4) financial health base upon adjusted operating profit margin, and (5) community health survey performance based upon the Bloomberg American Health Initiative and the Center for Health Equity at the Johns Hopkins Bloomberg School of Public Health. The three criteria are (1) does the hospital provide services critical to the community's health and offer critical preventive services; (2) do they partner with local organizations to implement critical care programs; and (3) do they act as an anchor institution by contributing to local economic and social progress.

For the top one hundred, only four of the classes made the final selection (Figure 2). This sample of hospitals was selected for this study since it represents a diversified selection by being different-sized hospitals for comparison. Further, the selection criteria provided standardization and control for a sample to study that varied on a single parameter—size.

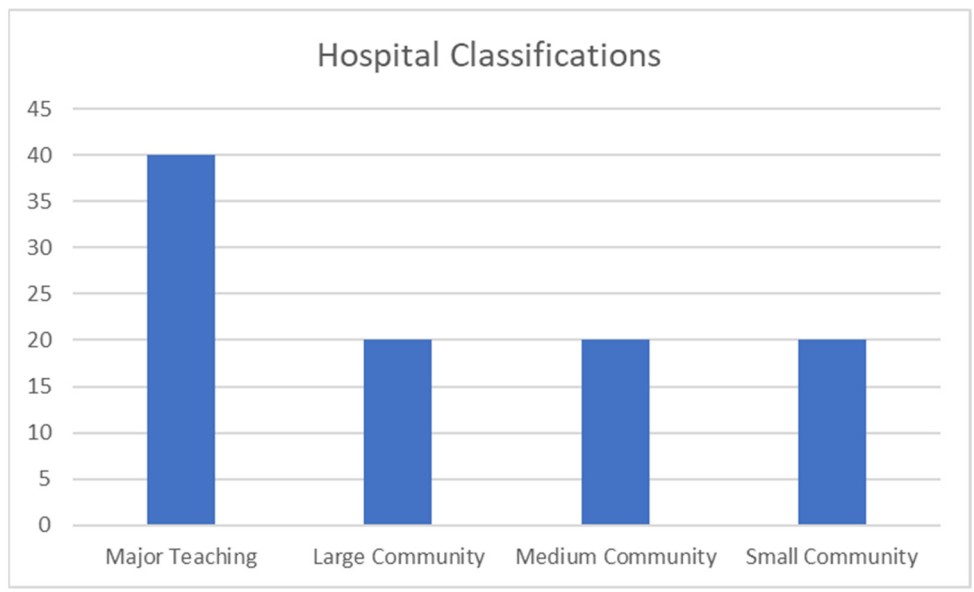

**Figure 2.** Breakdown of the top one hundred hospitals by classification.

*4.2. Investigation*

The assessment was conducted by visiting all one hundred hospital websites. A manual review was performed to assess the website design for discovery and presentation of the privacy notification. The assessment included coverage by quantifying the questions as follows:

1. Location of the initial entry to the privacy notice is on the first page of the hospital website (Yes/No).
2. Location on the website page of the initial entry to the privacy notice (Top, Bottom, Left, Right).
3. Capture and tally the terms used for the title for the initial entry to the privacy notice.
4. Capture and tally the number of URL clicks to get to the privacy notice.
5. Capture and tally the use of one or more of the following attention-eliciting website design elements: drag, dropdown, swipe, animation, or checkboxes.
6. Compare and contrast the website design differences between "major" teaching hospitals, top large community hospitals, top medium community hospitals, and top small community hospitals based upon (1) through (5).

## 5. Results

### 5.1. Location of the Initial Entry to the Privacy Notice (Figure 3)

Question 1. Is the initial entry to the privacy notice located on the first page of the hospital website?

In Figure 3, ninety-five (95%) of the websites had the initial entry point to the privacy policy or statement document on the first page of the website. Across the hospital types, the average was 96% had the initial entry on the first page with a standard deviation of 4% and a coefficient of variance of 5.39%. Of the five that did not have an initial entry on the first page, three were major teaching hospitals, and two were small community hospitals. As described prior, it is important for legal agreements like a privacy policy to be accessible. If they are not, then the legality of the terms may not be enforceable. By being available on the first page for consistency, a person going to the hospital website can assume there is a link to the privacy policy on the first landing page.

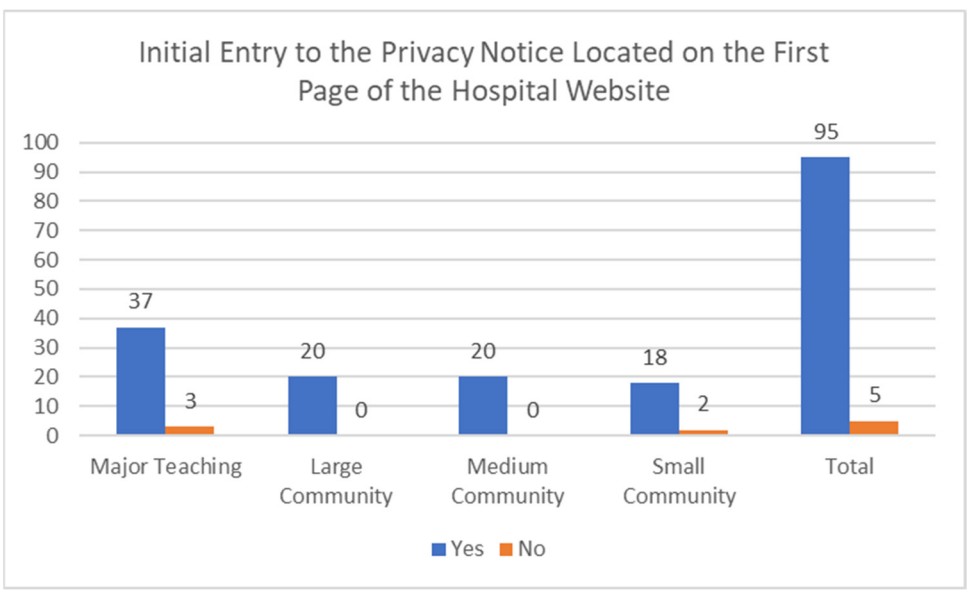

**Figure 3.** Initial Entry to the Privacy Notice.

### 5.2. Location on the Website Page of the Initial Entry to the Privacy Notice

Question 2. Where on the website page is the initial entry to the privacy notice located?

Ninety-five of the initial entry points to the privacy notice were at the bottom of the website's main page. Across the hospital types, the average was 96% of those that had the location's initial entry on the bottom of the first page, with a standard deviation of 4% and a coefficient of variance of 5.39%. Three of the major teaching hospitals and two of the small community hospitals did not have initial entry points on the main page (Figure 4). An area for investigation would be why is the industry trend to put it at the bottom of the page. It is important from an accessibility perspective and for attention gaining to have it available as quickly as possible, which would logically point to a top-level point of reference. Schaub et al. [15] examined "why" privacy notices failed to inform end-users of their rights. They found that the design process needs to include timing as to when the notice should be presented to the user, how it is delivered, how it is displayed or communicated, and what end-user choices are integrated into the notice. One problem they noted was the location of the notice is usually not readily accessible, i.e., having to go to a website, follow menus, contact the business.

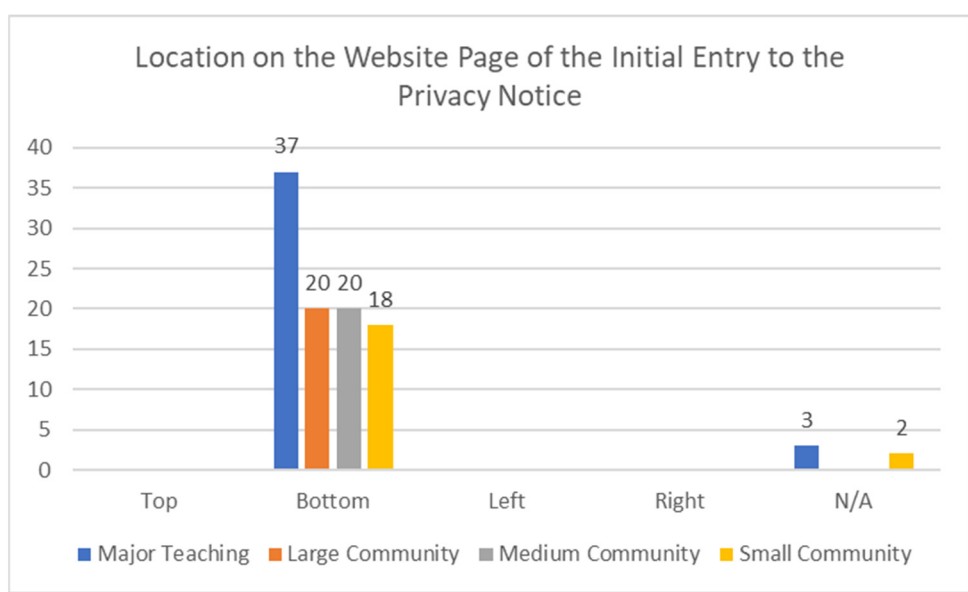

**Figure 4.** Location on the Website page of the Initial Entry.

*5.3. Term(s) Used for the Title of the Initial Entry to the Privacy Notice*

Question 3. What is/are the terms used for the title for the initial entry to the privacy notice?

On average, 85% of the hospitals had the word "privacy" in the title, with a standard deviation of 7% and a coefficient of 8%. The top initial entry terminology to reach the privacy policy was "privacy policy". "Privacy policy" was the top phrase across all four hospital classifications. The second highly used phrase was "Notice of Privacy Practice(s) (23%) for all four hospital types. The third most used phrase was "Privacy" (11%) by all. Other phrases that were used included "Privacy Practices", "Privacy Statement", and "Patient Privacy Practices". All four hospital classifications had other unique phrases. However, they usually contained the word "Privacy", "Legal", or "Policy" in them. Five of the sites did not have an initial entry point, thus no phrase for locating the privacy policy (Figure 5). Since the user is locating the privacy policy and it is important to gain their attention, having the word privacy in the initial entry is a logical design aspect.

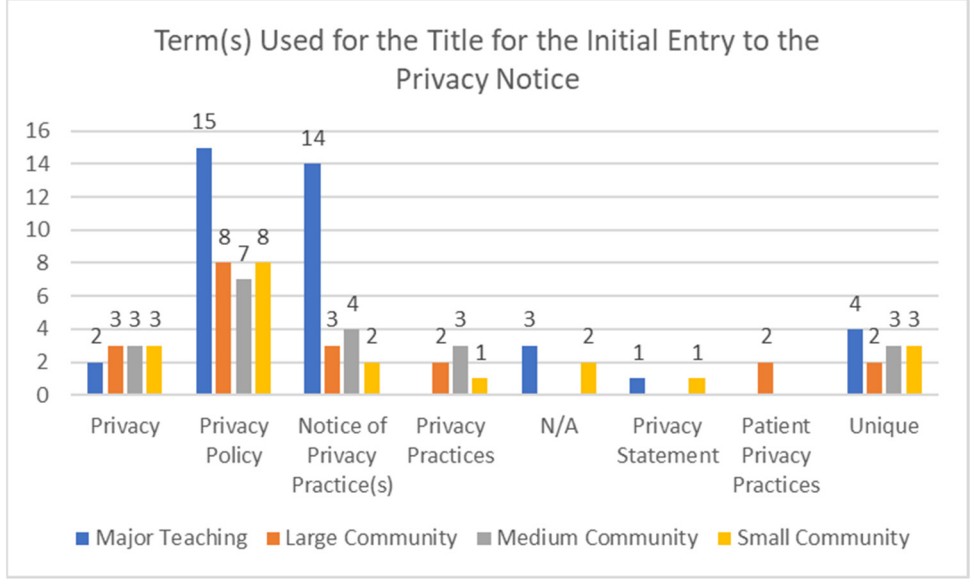

**Figure 5.** Term Used for the Title for the Initial Entry.

### 5.4. Number of URL Clicks to Get to the Privacy Notice

Question 4. How many URL clicks does it take to get to the privacy notice?

On average, 91% used one to two clicks with a standard deviation of 8% and a coefficient of 8%. From the initial entry phrase, fifty-three of the hospitals required one click on the URL to reach the privacy policy. Thirty-seven required two URL clicks. Five required three clicks. None of the websites required more than three URL clicks (Figure 6). Five of the hospitals did not have a privacy policy locatable. One of the major teaching hospitals' links did not work. Since the word "Privacy" appeared in all but three, a user had discernible information to identify where the hospital privacy was located. Ideally, anything more than a click depth of three should not be used. Thus, this sample did better with using only one to two URL levels to access the privacy policy.

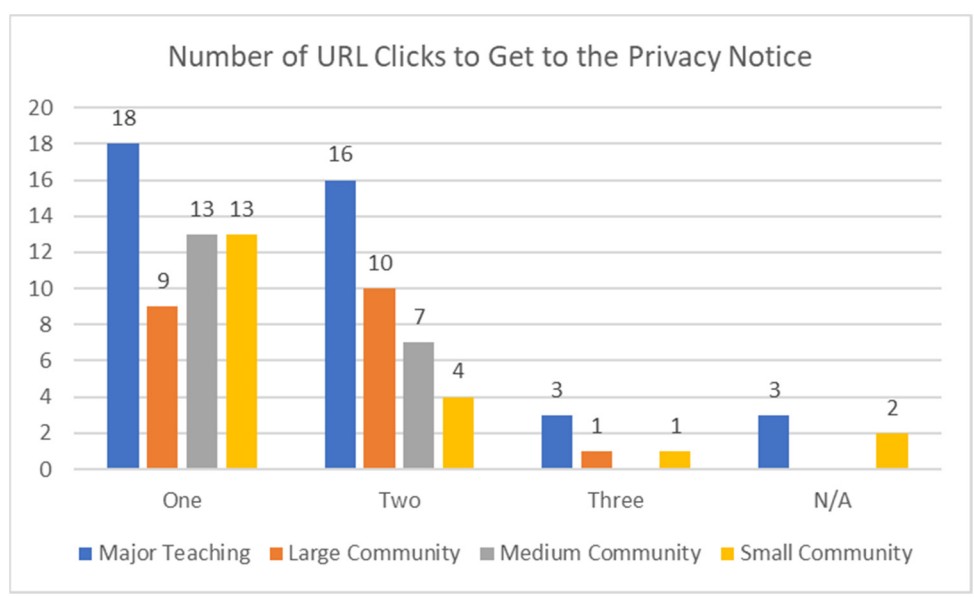

**Figure 6.** Number of URL Clicks.

### 5.5. Use of One or More of the Attention-Eliciting Website Design Elements

Question 5. Is any attention-eliciting website design elements, as cited in research, used such as drag, dropdowns, swipe, animation, or checkboxes?

Plain text was primarily used to navigate the user to the location of the privacy policy. Across the hospital types, the average was 96% used plaintext with a standard deviation of 4% and a coefficient of variance of 5.39%. There were thirteen instances of using a menu pull-down in conjunction with the plain text to lead the user to the privacy policy. Each of these pull-down choices led to a specific facility location that then had its privacy policy posted on its website. None of the hospital classifications used any alternate attention-gaining website design practice that has been cited in the research literature as recommendations to obtain user compliance to read and review a privacy policy (Figure 7). Karegar et al. [16] measured user engagement using different formats than the normal text found in notices. They used swipe, drag, and drop, and checkboxes. Their study revealed that creating different ways to interact with the form could increase user attention to sections of it. To increase user attention to review a privacy policy, Tabassum et al. [17] used the novel approach of a comic-based interface. Their results showed that user attention was held longer than in text-based formats.

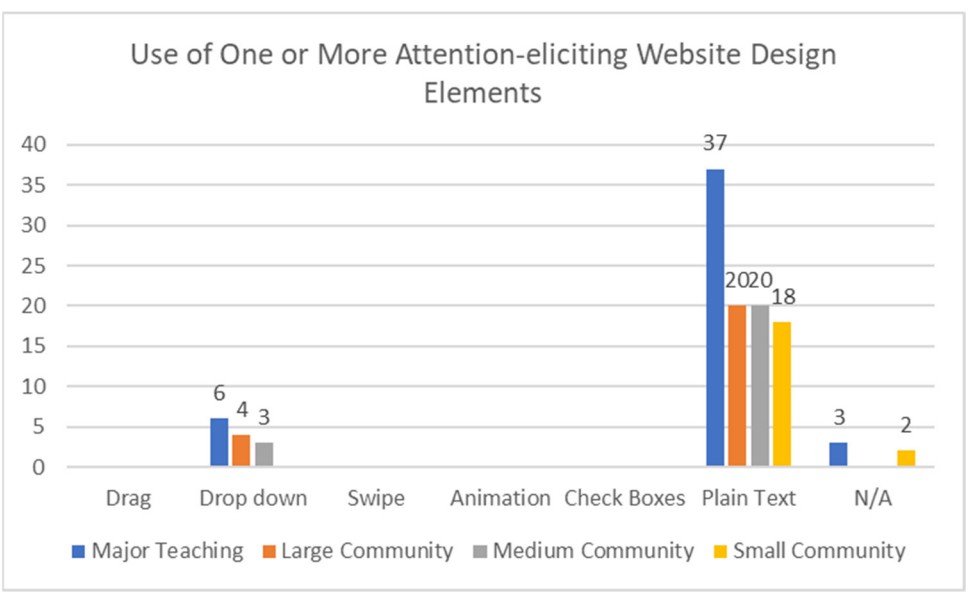

**Figure 7.** Use of Attention-eliciting Website Design Elements.

*5.6. Compare and Contrast of Hospital Class Types*

Question 6. Are there website design differences between "major" teaching hospitals, top large community hospitals, top medium community hospitals, and top small community hospitals?

There were many similarities across all the hospital classifications. First, all had the initial entry to their privacy policy on their first website page. The only exception was five hospitals that did not have a privacy policy that could be located. This occurred with three major teaching and two small community hospital websites. Second, the locations of the initial entry for all except the same five mentioned previously were located at the bottom of the first page. Third, all but seventeen of the initial entry titles used the word "privacy" in the title. All the classifications had a few websites that used a "unique" initial entry worded title. Over half of the hospitals required a single URL click to get to the privacy policy. The max number of URL clicks was three to get to the policy. All the hospitals primarily used text instead of an alternative attention-eliciting website design element. Thirteen did combine their text with the use of a pull-down menu to lead the user to a particular facility or department location to obtain access to the privacy policy.

The differences among the hospital classification were fewer than the similarities. The small community hospitals used 100% text and no other attention-eliciting website design elements. The five hospitals that did not have their initial entry point to their privacy policy included the major teaching and small community hospitals. When it came to the title terminology for the initial entry, the only classification to use "Patient Privacy Practices" were large community hospitals. The title "Privacy Practices" was not used by major teaching hospitals, and the other categories did use it. Only medium community hospitals did not require any more than two URL clicks to get to the privacy policy.

## 6. Discussion

Research representing cross-global industries has identified the need to improve a user review of privacy policies. They include techniques and frameworks that can be used in website design to improve end-user evaluation. However, there has not been a study that investigated both. In this study, there was low variance in how hospital websites were designed to access their privacy policy. Of the ninety-five that had a privacy policy available, all had their initial entry URL on the first page and at the bottom. Moreover, all predominately used text instead of any attention-eliciting design elements. On average,

85% had the word "privacy" in the initial entry URL title. None of the hospitals required more than three URL clicks to get to the privacy policy.

The observed consistency in design is beneficial for frequent visitors to hospital websites when a client wishes to be informed of their privacy rights. This facilitates the ability to locate it rapidly. Having the initial entry at the bottom of the page and requiring up to three URL clicks to locate it could be challenging for users with disabilities that may have to rely on assistive devices to scan and read the website. Further, the lack of minimal use of attention-eliciting design elements was another call out because of the clientele that use hospitals. People with medical conditions may not have the mental capacity or could easily become fatigued. Spending time and energy to transverse a website and clicking URLs may not be an easy accomplishment when compared to a healthy individual.

For this sample, the near homogeneity of hospital website design demonstrates a perceived standard based upon an industry practice rather than the integration of techniques from research.

The framework proposed by this study provides a first step in assessing in a systematic manner the design of websites for privacy policy location design.

## 7. Limitations

This study's sample selection focused solely on the top 100 US hospital websites deemed to be the top per IBM/Fortune Magazine. The quantitative measures were verified manually and not with the use of a computer algorithm that could control bias. Results were not confirmed or compared to a person with disabilities experience. Five of the websites did not have a "locatable" privacy policy.

## 8. Future Research

Research for designing websites to enhance the ability to communicate to a person about their privacy rights is limited. WCAG guidelines focus on accessibility design, but there is an opportunity to explore additional design techniques to expand into privacy notification. This study's research could be replicated to use a more diverse group of investigators or to conduct focus groups with persons with diverse backgrounds. With the limited availability of research on privacy notification to the disabled, there are a variety of opportunities for future research. More accessibility tools for reading websites like puff and sip can be explored. Moreover, there is an opportunity to include different manufacturers of those tools to compare. A global approach could be launched to learn an expanded set of findings on similarities and differences in the design of websites for communicating privacy rights in other countries. This would be interesting because of the tighter legal responsibilities with GDPR. China has introduced stricter information privacy laws. More business and institution websites should be included. Another opportunity exists with creating test websites designed with research-recommended attention-eliciting techniques and comparing them to the popularity of using predominantly text based with various focus groups. This study could be extended to compare hospitals that are not in the top 100. These results can carry over to creating standards needed to improve website design for privacy notification review. More research for designing websites to enhance privacy policy location and policy design together is needed.

## 9. Conclusions

This research brings focus to the website design techniques for privacy notification by proposing a standardized framework for assessment. The current literature research focuses primarily on the privacy policy and its content. Research on healthcare websites focuses on their design and ease of use. A gap in research exists for assessing website design for supporting findings and locating a privacy policy on a healthcare website. In general, there is an opportunity to examine website design techniques for more than just healthcare to create greater transparency and ease of locating privacy policies. The study reveals website design based on hospital industry trends. It brings to light strong similarities of

using text-based, first-page, and bottom-location design to facilitate or not easily facilitate the location of the privacy policy, depending upon the user's perspective. It highlights a framework assessment for webpage design for privacy policy location to standardize the collection and baseline of an industry trend. The importance of privacy communication is a growing global concern as more and more data are collected, processed, and used. Few subjects take the time or have the liberty to understand their privacy rights completely. There should be more attention by the collector and processor of privacy data to improve the website design to communicate a privacy policy.

**Author Contributions:** Conceptualization, K.S.; methodology, K.S.; software, K.S.; validation, K.S.; formal analysis, K.S.; investigation, K.S.; resources, K.S.; data curation, K.S.; writing—original draft preparation, K.S.; writing—review and editing, K.S. and M.S.; visualization, K.S.; supervision, K.R. and M.S.; project administration, K.S.; funding acquisition, Not applicable. All authors have read and agreed to the published version of the manuscript.

**Funding:** This research received no external funding.

**Data Availability Statement:** No new data were created.

**Conflicts of Interest:** The authors declare no conflict of interest.

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
