# Peer review of "A Descriptive Study of Webpage Designs for Posting Privacy Policies for Different-Sized US Hospitals to Create an Assessment Framework"

_futureinternet, doi:10.3390/fi15030112_

Round 1

Reviewer 1 Report

Summary: " This study investigates various hospital website design practices to develop a foundation framework using WGAC principles for assessment to identify techniques in publishing privacy policies. Based upon select WGAC principles, a framework for assessing website design to support privacy policy posting was implemented to compare the website design of different-sized hospitals."

Major Comments:

  1. Authors should explicitly specify the novelty of their work. What progress against the most recent state-of-the-art similar studies was made?
  2. Conclusions should be amended to incorporate a broader discussion of this specific study's significance and potential application.
  3. What kind of similar solutions are available to solve privacy issues on websites? The related work needs to be included. 
  4. English throughout the manuscript needs to be improved.

Author Response

Thank-you for your valuable time and feedback! Please see attached updated manuscript and responses below:

Major Comments:

  1. Authors should explicitly specify the novelty of their work. What progress against the most recent state-of-the-art similar studies was made?

In the healthcare industry, where sensitive data is being collected, research on how a user navigates to privacy policies for different size hospital websites is limited. Studies exist about privacy policies or website design and not both. This descriptive study involved ascertaining commonalities and differences amongst different sized hospitals website designs for supporting privacy policies. A foundation framework was created using Web Content Accessibility Guidelines (WGAC) principles and literature review findings for evaluating practices for website publishing of privacy policies. 

  1. Conclusions should be amended to incorporate a broader discussion of this specific study's significance and potential application.

Additional information was provided:

Conclusion

This research brings focus on the website design techniques for privacy notification by proposing a standardized framework for assessment. Current literature research focuses primarily on the privacy policy and its content. Research on healthcare websites focuses on the design and ease of use. A gap in research exists for assessing website design for supporting finding and locating a privacy policy on a healthcare website. In general, there is opportunity to examine website design techniques for more than just healthcare for creating greater transparency and ease to locate privacy policies. The study reveals website design based upon hospital industry trends. It brings to light strong similarities of using text based, first page and bottom location design to facilitate or not easily facilitate the location of the privacy policy depending upon the users perspective. It highlights a framework assessment for webpage design for privacy policy location to standardize the collection and baseline of an industry trend. The importance of privacy communication is a growing, global concern as more and more data is collected, processed, and used. Few subjects take the time or have the liberty to completely understand what their privacy rights are. There should be more attention by the collector and processor of privacy data to improve the website design to communicate a privacy policy.

  1. What kind of similar solutions are available to solve privacy issues on websites? The related work needs to be included. Additional information was provided:

Privacy Policy Publishing Assessments

Health information technologies (HIT) are important for providing users information. This includes websites. LaMonica, et al. [18] performed research that identified a growing concern to understand privacy risks associated with HIT and the confusing nature of privacy policies. The study applied a 23-item privacy policy risk assessment tool to assess whether HITs supported recommended privacy policy standards. Results found that users “wanted privacy information to be easily accessible, transparent, and user-friendly”. Zimmeck, et al. [19] developed a proof-of-concept browser extension that analyzed privacy policies for essential policy terms. The Privee tool architecture was based upon the notice-and-choice principle for locating and understanding privacy policies on websites. It increased understanding and transparency by extracting key terms from privacy policies. Johnson, et al. [20] researched templates for authoring security and privacy policies. The framework used new and existing policy guidelines to develop standardized systems. These assessments and frameworks focused on privacy policies and content rather than including design concepts of the website. These studies highlight the limited research on designing websites for locating the privacy policy.

English throughout the manuscript needs to be improved. Spell checker and grammar review performed. Further, proofreading was performed after the changes and additions.

Reviewer 2 Report

This article proposes an assessment framework for posting privacy policies focusing on US hospital website design.

Furthermore, the following points regarding motivation and argumentation should be considered in the revision of this article:

- for reading purposes, I would suggest to construct the storyline of the Abstract without using references.
- in the Introduction the motivation behind choosing to focus on the health industry, and in particular, on hospital data, should be discussed.
- in the Background section, it would be good to discuss what other similar assessment frameworks have been proposed in the field of cyber security.
- in the Methodology section, the criteria established for manually reviewing the websites should be motivated.

Author Response

Thank-you for your valuable time and feedback! Please see attached updated manuscript and responses below:

Furthermore, the following points regarding motivation and argumentation should be considered in the revision of this article:

  • for reading purposes, I would suggest to construct the storyline of the Abstract without using references. You suggestions were applied to the Abstract.
    - in the Introduction the motivation behind choosing to focus on the health industry, and in particular, on hospital data, should be discussed. This section was reworked to discuss healthcare websites.
    - in the Background section, it would be good to discuss what other similar assessment frameworks have been proposed in the field of cyber security. Added the following:
  • 2.5 Privacy Policy Publishing Assessments

    Health information technologies (HIT) are important for providing users information. This includes websites. LaMonica, et al. [18] performed research that identified a growing concern to understand privacy risks associated with HIT and the confusing nature of privacy policies. The study applied a 23-item privacy policy risk assessment tool to assess whether HITs supported recommended privacy policy standards. Results found that users “wanted privacy information to be easily accessible, transparent, and user-friendly”. Zimmeck, et al. [19] developed a proof-of-concept browser extension that analyzed privacy policies for essential policy terms. The Privee tool architecture was based upon the notice-and-choice principle for locating and understanding privacy policies on websites. It increased understanding and transparency by extracting key terms from privacy policies. Johnson, et al. [20] researched templates for authoring security and privacy policies. The framework used new and existing policy guidelines to develop standardized systems. These assessments and frameworks focused on privacy policies and content rather than including design concepts of the website. These studies highlight the limited research on designing websites for locating the privacy policy.

    - in the Methodology section, the criteria established for manually reviewing the websites should be motivated. Added the following:
  • Manual review was chosen since no tools exist to perform such an assessment. Automating would require creating different scripts for parsing the unstructured text. Depending on the website front end design, there are challenges for creating a script to record the depth of the URLs to locate the privacy policy. The policy could be a PDF, HTML code, or various other applications could have been used. 

Reviewer 3 Report

-The paper presents a descriptive study that ascertained commonalities and differences amongst different sized hospitals website designs to support privacy policies.

-When listing items, I think it is clearer to the reader if you use open and close parenthesis, for example: "The main contributions of this study are as follows: (1) The literature review...".

-What was the rationale of selecting hospital websites from the list of the "2021 Fortune/IBM Health 100 Top Hospitals"? There is no clear justification on why you used that list.

-If you refer to specific figures or tables, make sure to put the first letter in capital: "Figure 1".

-Was it necessary to include a figure to illustrate the count of URL clicks to access the detailed privacy notice or statement of a website? I think the statement that you included would have been enough.

-Are the questions considered in your investigation enough to provide relevant results? I think you should have added more relevant questions to provide more insightful results.

-While the webpage design is important in how the privacy policies are presented to the user, I think it would be difficult for the healthcare providers (public and private) to agree on a standardised design or way in which the privacy policies are presented to the user. Furthermore, you are only considering the top 100 hospitals in the United States, have you considered the case of hospitals in the United States that are not in the top 100? What about hospitals around the world? I think your research presents an initial step, but you should consider other cases in your future work.

Author Response

First, I want to thank you for your time and feedback. I have uploaded an updated version of my manuscript with your suggested changes. Also, I have responded point by point to your comments.

-The paper presents a descriptive study that ascertained commonalities and differences amongst different sized hospitals website designs to support privacy policies.

-When listing items, I think it is clearer to the reader if you use open and close parenthesis, for example: "The main contributions of this study are as follows: (1) The literature review...". These were all searched for and changed.

-What was the rationale of selecting hospital websites from the list of the "2021 Fortune/IBM Health 100 Top Hospitals"? There is no clear justification on why you used that list. This sample of hospitals was selected for this study since it represent a diversified selection by being different sized hospitals for comparison. Further, the selection criteria provided standardization and control for a sample to study that varied on a single parameter – size.

-If you refer to specific figures or tables, make sure to put the first letter in capital: "Figure 1". A search was done to ensure this was addressed.

-Was it necessary to include a figure to illustrate the count of URL clicks to access the detailed privacy notice or statement of a website? I think the statement that you included would have been enough.  For the reader, an illustration showing the comparisons I felt provided the visual reviewer consistency on how I presented my findings for each question. Thank-you for a providing this!

-Are the questions considered in your investigation enough to provide relevant results? I think you should have added more relevant questions to provide more insightful results. This is a very good point. Since there is limited research that reviews both the website design together with focusing on locating the privacy policy, I used WCAG guidelines integrated with the findings from my literature search. Research either looks at website design OR privacy policies and not both. There is a gap in research to have a tool to assess/audit website designs to help find a privacy policy to increase reviews by users.

-While the webpage design is important in how the privacy policies are presented to the user, I think it would be difficult for the healthcare providers (public and private) to agree on a standardised design or way in which the privacy policies are presented to the user. From a legal perspective, the policy needs to be reviewed to be binding. If there is a better method for presenting the policy to ensure this, a trend to standardize could be started. Furthermore, you are only considering the top 100 hospitals in the United States, have you considered the case of hospitals in the United States that are not in the top 100? What about hospitals around the world? I think your research presents an initial step, but you should consider other cases in your future work. I had mentioned a "global" opportunity. But, I did reword it and added about comparison of non-top 100. A global approach could be launched to learn an expanded set of learnings on similarities and differences in the design of websites for communicating privacy rights in other countries. This would be interesting because of the tighter legal responsibilities with GDPR. China has introduced stricter information privacy laws. More business and institution websites should be included. Another opportunity exists with creating test websites designed with research recommended attention-eliciting techniques and comparing them to the popularity of using predominantly text based with various focus groups. This study could be extended to compare hospitals that are not the top 100.

Reviewer 4 Report

With more and more data being collected, processed, and used for profit, surveillance, and population control, the importance of private communications is of increasing global concern. Few people take the time to understand the integrity of their privacy. This paper focuses on web design techniques for privacy notices by proposing a standardized evaluation framework. The findings reveal trends in website design based on hospital industry trends. It reveals strong similarities in the use of the text-based first page and bottom position design and the location of privacy policies based on the user's perspective of promotion or non-promotion. It highlights a framework for evaluating privacy policy location web design to standardize the collection and baseline of industry trends. Privacy data collectors and processors should pay more attention to improving the design of their websites to communicate privacy policies. Indeed, the idea is interesting. Thus, I can recommend this manuscript for publication in Future Internet.

Author Response

Thank-you for your valuable feedback and time! Please see attached updated manuscript.

Round 2

Reviewer 1 Report

The authors have addressed all the comments. The paper can be accepted in its present form.